# Student engagement and attendance are central mechanisms interacting with inclusive and equitable quality education: Evidence from Afghanistan and Pakistan

Jean-Francois Trani [1,2,3¤a]*, Robbie Hart [4¤b]

1 Brown School, Washington University in St Louis, St. Louis, Missouri, United States of America, 2 National Center for Arts and Crafts, Paris, France, 3 Department of Psychology, University of Johannesburg, Johannesburg, South Africa, 4 William L. Brown Center, Missouri Botanical Garden and Washington University in St Louis, St. Louis, Missouri, United States of America

¤a Current address: One Brookings Drive, St. Louis, Missouri, United States of America
¤b Current address: William L. Brown Center, Missouri Botanical Garden, St. Louis, Missouri, United States of America
* jtrani@wustl.edu

**Data Availability Statement:** The original dataset for this study is available on the UK data service

## Abstract

Despite considerable progress in the field of education science, there is currently no consensus on the components that define inclusive and equitable quality education, how they are articulated with one another, and what are the best interventions to foster inclusive and equitable quality education. Current research investigates separately components of what appears to be a complex dynamic system with feedback interactions. To characterize this system and identify structures that encompass inclusive and equitable quality education, we used a community-based system dynamics approach. This approach hypothesizes that community perceptions of the local school system is essential to define it. We therefore conducted 648 participatory Group Model Building workshops with school stakeholders (children, teachers, parents and members of school management committees) resulting in as many models in 99 schools of Afghanistan and Pakistan. To identify common components across models built by participants in two waves of schools' model building workshops, we applied techniques from multivariate analysis of ecological communities. Even across wide differences in participants' situations and roles in the educational process, their models expressed a common reinforcing feedback loop which connected child inclusive and equitable quality education to two other components: 1) child engagement in and motivation for education, and 2) child attendance. Increases in any of these three components were perceived to drive increases in the others. We also found that child focus on learning was commonly expressed as interacting with this generic structure. Any educational reform should simultaneously and primarily embrace learners' diversity, combine policy principles of ensuring easy equitable access to foster attendance, and promote student interest and engagement in learning through child centered pedagogy and non-discriminatory teaching practices while giving school communities power for implementation.

website: https://reshare.ukdataservice.ac.uk/856670/.

**Funding:** The study was funded by a grant from the Economic and Social Research Council of the United Kingdom (grant number ES/P005799/1). The funders had no role in study design, data collection and analysis, decision to publish, or preparation of the manuscript.

**Competing interests:** The authors have declared that no competing interests exist.

## Introduction

Understanding child learning processes to improve both cognitive and non-cognitive learning outcomes and promote overall inclusive and equitable quality education has been a central focus of multiple disciplines from biology and neurosciences to economics, psychology and sociology and, more recently, the field of learning and development [1]. This field prioritizes a holistic approach to education that conceptualizes the teachers (virtues, qualities and professional skills) [2–4], the school (organization and infrastructure) and its environment (student-centered instructional strategies, student cognitive, social and emotional support mechanisms, community and family involvement outcomes) [5], as a system in which overall child development occurs. If there is not, to date, a consensus on what inclusive and equitable quality education embraces and how to prioritize policies and programs, there seems to be some agreement on a goal of providing instruction for all children to prepare them for a sustainable future life. All children should be present, participate and learn in school [6]. Values of fairness, justice and democracy underpin quality education for all [7, 8]. Achieving the democratic goal of forming future critical and knowledgeable citizens of a country is an enduring battle that require innovative approaches [9]. This task is even harder to achieve in an increasingly unfair world where a few concentrate privileges and wealth [10]. Focusing on social justice, human development and the capabilities approach, and recognizing not just the instrumental value of education but also its inherent value [11, 12], our study investigates how participatory methods allow to shift away from a right to education towards a right to quality and equity in learning for all children, including the most disadvantaged students, namely children with disabilities, children in remote rural areas and from poor families, girls and ethnic minorities who have lower enrolment rates, higher rates of repetition and drop out before completion, and lower academic achievement [13–15]. This conception promotes the idea of an inclusive system of education as one that does not simply welcome disadvantaged students and fight discrimination, but also one that aims at meeting the social and academic needs of all students and fostering a sense of community where all of them feel they belong and learn effectively [16].

Recognizing this, the fourth United Nations Sustainable Development Goal (SDG4) has formalized and created a new impetus towards inclusive and equitable quality education for all [17]. In low and middle income countries (LMICs), including in Afghanistan and Pakistan–two countries characterized by low attendance and high dropout rates as well as limited academic achievement–multiple theories have been tested to promote child enrolment, attendance and learning, but have been essentially focused on cognitive learning outcomes measured by test scores [18] to the exclusion of non-cognitive outcomes or life skills [19]. Policies and programs–often imposed by the central government with little local input–have focused on different dimensions, either on the demand side that focus on households (e.g. conditional cash transfer programs), or on school input (e.g. access through building, latrines, water; learning material, teacher salary, training and initial level of education) [20]. Overall, interventions to promote inclusive and equitable quality education in LMICs have mixed results. Interventions promoting smaller class sizes, teacher incentives, learning materials, and cash transfers have been found to have short-term or demographically limited impacts and more on attendance than on learning outcomes [21]. On the other hand, some one-factor interventions focusing on family demand for schooling (adult information and literacy programs), school inputs (building schools), pedagogy or school governance have shown better effectiveness in improving enrolment, attendance and learning outcomes [22]. One reason for these divergent results may be that most existing interventions focus on one-factor linear causal models to understand what appears to be a systemic, wicked problem [23]. If education represents a complex system [24], it may simply not be tractable with such approaches, and a

more comprehensive understanding is to be gained by methods that recognize multiple factors and the feedbacks among them as constituting the teaching and learning process.

Across a wide range of fields, local perceptions have been recognized as important to comprehensively understand complex problems. The understanding that stakeholders at the local level are both direct and long-term observers of the systems in which they operate, as well as change agents within them, has driven both pragmatic and philosophical arguments for inclusion of these perspectives in approaches to such problems as adaptation to climate change [25], natural resource overharvest [26], development of wellbeing (5), empowerment of women (6), and understanding oppression (7). However, a distinct challenge in the analysis of local perception is aggregation across myriad complex, qualitative stakeholder perspectives to reveal generic system structures. That is, among a diversity of perspectives and local contexts, what are the components and interactions upon which all stakeholders agree? To answer this question about the system surrounding inclusive and equitable quality education, we used a two-step process.

First, group model building allowed participants at a local level to conceptualize the education system in comparable terms. Group model building is a community-based systems dynamic approach in which a group of stakeholders and researchers co-design mental models called causal loop diagrams (CLDs) using shared language, generate a common understanding of a wicked problem and identify and test shared innovative practical solutions to address this problem [27, 28]. Using this standardized community-based method allowed us to leverage responses from diverse respondents in rural Afghanistan and rural Pakistan. Both countries have been facing the adverse effects of the COVID-19 pandemic which has exacerbated pre-existing inequities, particularly for vulnerable children. Even before the pandemic, both countries were characterized by alarming education indicators: 40%, or 3.7 million in Afghanistan [29], and one-third, or 22.5 million in Pakistan [30] of primary school age children—mostly girls—were out of school. The mean years of schooling were only 3.2 years in Afghanistan and 4.7 years in Pakistan. Vulnerable children have lower enrolment rates and are more likely to repeat grades or drop out before completion [31–36]. Nevertheless, both countries have committed to achieve SDG4 and "Education For All" [37], providing inclusive and equitable quality education, with emphasis on vulnerable children [31, 32]. Although both countries have developed a range of policy and practice initiatives, actual improvements have been limited, and evidence-based implementation strategies are lacking [38–40]. In Afghanistan, the return to power of the Taliban movement further threatens past commitment to inclusive and equitable quality education.

As a second step, to solve the problem of identifying commonalities across many diverse qualitative perceptions, we applied techniques from multivariate analysis of ecological communities [41, 42], previously used in other local perception domains [43]. After identifying the highest-consensus CLD components and interactions among them (feedback loops by which components are either reinforced or balanced), we used these ecological methods to quantitatively test similarities or differences in the systems expressed by stakeholders: 1) of differing roles in the education system (students, teachers, parents, and school management committees' members), 2) in different geographical contexts, 3) difference in wealth, 4) for both sex and 5) at two time periods (before and after an intervention decided by consensus in each school).

An understanding of connections between the school environment, the children's circumstances and characteristics, and inclusive and equitable quality education based on local perceptions can inform efforts to build policies and programs.

## Methods

We conducted two rounds of one day long group model building workshops–between October 2018 and April 2019 (n = 323) and between November 2019 and July 2020 (n = 325) respectively–in 40 schools of Badakhshan, Ghazni and Takhar provinces of Afghanistan and 59 schools of Punjab and Sindh provinces of Pakistan. Group model building is a set of Community Based System Dynamics (CBSD) methods that provides a structured process and forum for stakeholders to identify and prioritize issues through the language of systems [28, 44]. CBSD is a system science approach built on Community Based Participatory Research [45] that examines interactions between multiple factors and stakeholders who have a personal investment in a challenging problem [46, 47], referred to as *wicked* and *messy problems* [28]. CBSD promotes local ownership/leadership in solving community problems, provides new tools for practical problem-solving, builds capacity, and directly addresses underlying factors that impede equity between the community and researchers [48–50].

Scripts developed for the group model building workshops were tested with school stakeholders (students, teachers, parents, school management committees) from Kabul, Afghanistan and Rahim Yar Khan, Pakistan. School management committees (called in Afghanistan school *Shuras* or assemblies and in Pakistan Village Education Committees) are a consultative body bringing together selected parents, headmaster, teachers and village elders. In Pakistan, we combined teacher and school management committee groups because their membership overlapped.

We conducted four distinct activities: (i) focus group discussions (FGDs) to define a shared vision of inclusive and equitable quality education; (ii) variable elicitation to identify multiple factors perceived to influence inclusive and equitable quality education; (iii) connections circle to establish perceived connections between identified factors and identify causal feedback loops and, (iv) action ideas elicitation for participants to agree on interventions to affect the system. Trained facilitators assisted school stakeholders in the process. Visual causal maps (Fig 1) were elaborated by children, parents, teachers and school management committees. During each workshop, several iterations of the model with ongoing discussion of connections between factors were elaborated before a final visualization was agreed upon by the group. The building of the CLD and its review allows researchers to gauge each group of participants' consensus vision of the system.

CLDs apply simple rules. They build on causal relationships between two components (a *cause* and an *effect*). For example, many groups expressed that if *child attendance* increases, *child participation in the learning process* increases as a result (see Fig 1). The reverse is implied: if *child attendance* decreases (as was observed during the COVID19 pandemic), *child participation in the learning process* decreases. This is an example of a positive causal relationship where an increase in the *cause* drives an increase in the *effect* (indicated during model building by arrows with a plus sign); negative causal relationships are also possible (indicated during model building by arrows with a minus sign) when an increase in the *cause* drives a decrease in the *effect*. For instance, if *child attendance* increases, then *teacher scolding of children* decreases. Causal relationships may be directly expressed (*cause* increases *effect*), or downstream relationships may be implied by the system (*cause* increases *effect 1* which in turn increases *effect 2*). For example, the direct relationships i) *child health* increases *child attendance* and ii) *child attendance* increases *inclusion in the learning process* imply the downstream relationship iii) *child health* increases *inclusion in the learning process*, with *child attendance* as an intermediary component.

Because system thinking analyses the world through causal relationships in both directions, downstream causal relationships in a fully elaborated CLD will eventually form loops (*cause*

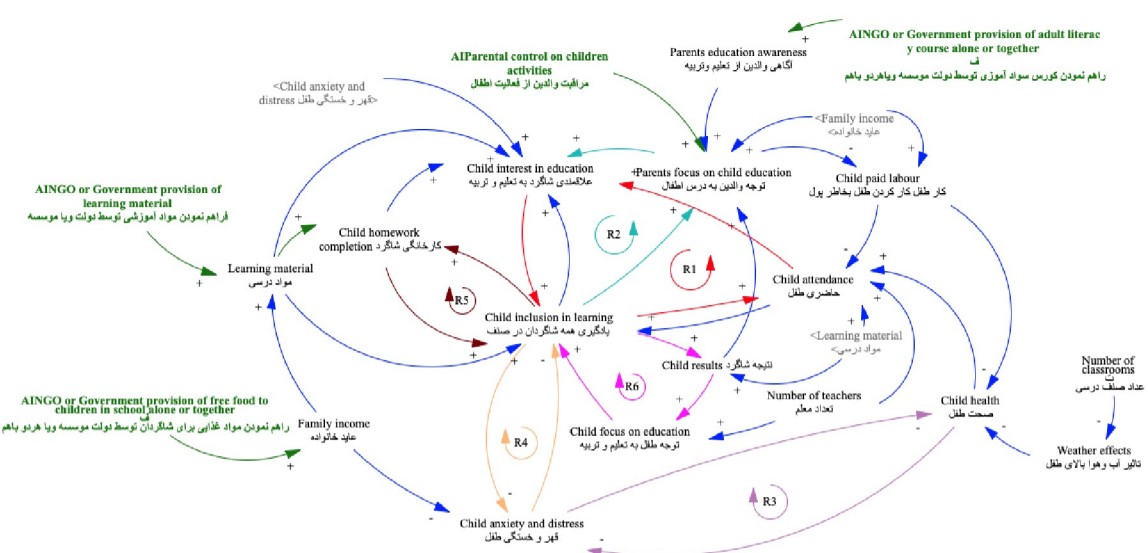

**Fig 1. Example of causal loop diagram approximately here.** Note: Causal loop diagram in Badakhshan, Afghanistan elaborated by students with text in English and Persian. R1 to R6: Causal reinforcing loops with R1 in red representing the generic structure; variables and connection in green represent action ideas which are interventions decided by children to improve the system.

increases *effect 1*…*increases *effect n* increases *cause*). There are two possible types of loops: reinforcing and balancing. In reinforcing loops (also called vicious or virtuous cycles) an initial increase in a cause will drive downstream increase in that cause and an initial decrease will drive downstream decrease, acting to amplify change in the system. In balancing loops, an initial increase in a cause will drive downstream decrease in that cause (and vice versa) acting to counteract change in the system [51].

After the conclusion of the workshops, facilitators reproduced the final Causal Loop Diagram models using Vensim® PLE software. Some revisions were introduced to clarify the logics of some causal links based on audio recording and notes taken by facilitators. Components were also translated, and terminology standardized, although broad categories were avoided, in order to retain specifics of local perception. We also uploaded the transcripts of activities into DeDoose® qualitative data analysis software, developed a coding structure and coded the workshop activities' discussions around themes relative to inclusive and equitable quality education following a deductive method [52]. New themes were added based on the existing literature. For the present paper, we used excerpts associated with the themes of interest.

We used techniques from multivariate analysis of ecological communities to compare identify features commonly perceived to be central to a successfully inclusive and equitable education system [41, 42]. These methods, which have been previously applied beyond ecology [53, 54], allowed us to compare CLDs by i) quantifying distance/similarity measures between them based on number of shared components, ii) distinguishing clusters of CLDs that are more similar to one another, and iii) aggregating across clusters to identify the common components shared by all. These multivariate analyses also allowed us to test which school, teacher or student characteristics correlate with differences among CLDs. To do this, we ordered CLDs built by each group using non-metric multi-dimensional scaling on Euclidian distance matrices of causal feedback loops expressed by each CLD using the R package vegan [42, 55]. We then projected the CLDs into an ordination space in which their distance from one another was defined by the difference of their causal links. Thus, CLDs that shared a great number of causal links

were plotted close together; those that differed in causal links were plotted farther apart. Finally, we tested how well the resulting distribution of CLDs in the ordination space was explained by five sets of characteristics of the groups who built them. First, we considered role in the educational system (children, teachers, school management committee members, parents). Second, the sex balance of the model-building group was considered, which we quantified as the percentage of participants identifying as male, or, for the sake of characterizing groups, a binning of this variable into models built by groups >80% female, >80% male, or mixed. Third, geography was examined at three nested levels: the 99 schools, eight administrative areas (within Afghanistan the provinces of Badakhshan, Takhar and Ghazni and the distinct district of Jaighori within Ghazni; within Pakistan the districts of Bahawalnagar, Rahim Yar Khan and Vehari in Punjab province and Gothki in Sindh province), and two countries (Afghanistan and Pakistan). Fourth, wealth was measured by two alternative indices with scores calculated using polychoric principal component analysis [56]. One characterized common durable goods: radios, mobile phones, TV sets, pressure cookers, lamps, refrigerators, generator/solar panels, sewing machines, bicycles, motorbikes/auto-rickshaws, cars, houses. The other characterized rural wealth: animals, i.e. camels, cows and buffalos, sheeps and goats and poultry. Each index was divided into three categories (lowest 20%/ middle 20–80%/ highest 20%). Fifth, time period, which captured the difference between workshops conducted before and workshops conducted after an intervention composed of an intensive two week inclusive education training and the implementations of three to five action ideas identified and selected by participants. Action ideas could include construction of infrastructure (such as classrooms, playground, restrooms, boundary walls, library), classroom material (chairs, tables, books) or procedures (regular parent-teacher meetings, classroom regulations, teacher and students' attitudes).

Ethical clearance was received from the Ministry of Education in Afghanistan, the National Rural Support Program Board in Pakistan and Washington University in St Louis, Human Research and Protection Office. We obtained informed verbal consent from participants in group model building workshops because no personal information was collected during the workshop.

## Results

### Diversity of local perception

We conducted 648 participatory Group Model Building workshops resulting in as many causal loop diagrams in 99 schools of Afghanistan and Pakistan to investigate what system structures characterize inclusive and equitable quality education. The more than 4000 participants in these model-building groups expressed 639 system components. Components ranged from school-internal, quantitative measures such as *child attendance* to external and qualitative such as *good home environment*, from general such as *teacher performance* to specific such as *teacher use of mobile phones*, and included components of human agency such as *child focus on education* as well as external environmental pressures such as *drought*. (We report standardized, translated components in italics here and a list of these and the ways they were expressed locally are available in Table 1).

The CLDs included a total of 6,057 unique direct causal links, with a mean of 50 direct links in each CLD ($25\%^{ile}$ 40, $75\%^{ile}$ 58). There were unique 54,372 downstream causal links–that is, links with one or more intermediary components–with a mean of 430 total causal links (both direct and downstream) in each CLD. The number of intermediary components in these causal links were generally few (mean 3, $25^{\%ile}$ 1, $75^{\%ile}$ 4). Fig 2 shows all causal links expressed as direct links by more than 25% of the CLDs (Fig 2A) or downstream links by more than 50%

**Table 1. Examples of standardized variables with definition and translation in local languages.**

| Standardized variable | Local definition | Urdu | Sindhi | Dari | Pashto |
|---|---|---|---|---|---|
| Child attendance | Children go regularly to school (instead of missing class because of household chores or working in the field or in any other activities or staying home doing nothing because of other impediment) | بچے کی حاضری | ٻار جي حاضري | حاضری طفل | د ماشوم حاضري |
| Good home environment | The home environment is conducive to education. Parents and children quarrels are limited. Children focus on their study (not on TV). There is no discrimination of parents between boys and girls, elders and youth. | گھر کا اچھا ماحول | گھر جو سٺو ماحول | محیط خوب فامیل | د کور ښه چاپيريال |
| Teacher performance | Teachers' pedagogy is effective and students get good results and learn well. Parents are satisfied and provide good feedback. A healthy competition developed between teachers. | قابل اور تجربہ کار استاد | قابل ۽ تجربيڪار استاد | فعالیت معلم | د ښوونکو فعالیت |
| Teachers' use of mobile phones | Teachers use electronic devices (mobile phones) in class which distracts them from paying attention to the students and their lesson. | استاد کا موبائل فون کا استعمال کرنا | استاد جي موبائل جو استعمال ڪرڻ | استفاده تیلفون توسط استاد در صنف | ښوونکي لخوا د ګرخنده تليفونونو کارول |
| Child focus on education | Children pay attention in class and are on top of things (homework, learning lessons). | بچے کی تعلیم پر توجہ | ٻار جي تعليم تي غور | توجه طفل به تعلیم و تربیه | د ماشومانو تمرکز پر زده کړي |
| Drought | Farming is affected by absence of rain and families' resources are strained by a drop in crops. | خشک سالی | ڊڪار | خشکی | وچکالي |
| Child engagement in and motivation for education | Child shows interest for the lesson and motivation for learning. | بچی پڑھا ئی میں دلچسپی | ٻار جي پڙھائي ۾ دلچسپي | علاقمندی شاگرد به تعلیم و تربیه | ماشومانو علاقه |
| Child inclusion and participation in the learning process | All student present in class actually developed learning skills and intervene during the lesson. | بچے کی پڑھائی میں شمولیت | ٻار جو پڙھڻ ۾ شمولیت | یادگیری همه شاگردان در صنف | په زده کړه کي د ماشومانو شاملول |

of the CLDs (Fig 2B). Only one negative link (*poverty* decreases *learning material*) met this criteria, all other links were positive. Implied causal links that were feedback loops (e.g., *inclusive and equitable quality education* increases *child homework completion* and *child homework completion* increases *inclusive and equitable quality education* imply the reinforcing feedback loop *inclusive and equitable quality education* increases *inclusive and equitable quality education*) were common and were among the most widely expressed links. Although the exact topology of CLDs varied, a central triangle–with each component reinforcing the other two–consistently emerged from the system expressed by the diverse model-building groups.

## A generic structure

Even across a wide range of local perceptions expressed in CLDs by participants across countries, languages, sex, socioeconomic strata, and roles in the educational system, CLDs commonly expressed a reinforcing feedback loop (Fig 3) which included three mutually reinforcing components: 1) child engagement in and motivation for education (below: *engagement*) 2) child attendance (below: *attendance*, see translation above), and 3) child inclusion and equitable participation in quality education (below: *inclusion*). The reinforcing causal links among these three components were the most widely reported of 60,419 causal links (Fig 3B) and were the only components expressed by >75% of CLDs.

Both within and across group characteristics (role in educational system, country, country subdivision, wealth, sex), and in both time periods, the components of this triangle had the highest consensus of any CLD components reported (Table 2). In addition, the most frequent causal links and the only causal links with >75% frequency were the reinforcing causal links of this generic structure (i.e., the positive influences of each component on the other two

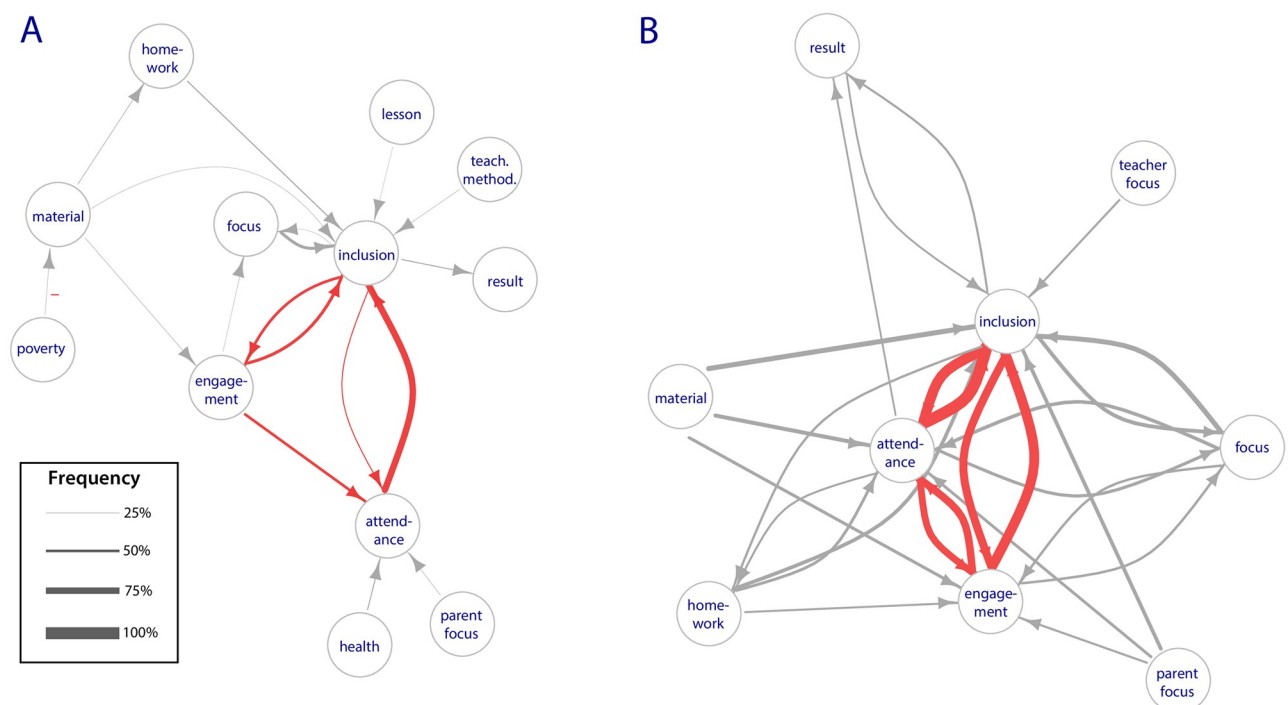

**Fig 2. Aggregate CLDs.** Note: Aggregated CLDs showing the most frequent causal links directly (A) or indirectly (B) expressed in CLDs built by 648 model-building groups of participants. All causal links expressed directly by more than 25% of the CLDs (in A) or indirectly by more than 50% CLDs (in B) are shown, with arrow weights proportional to the frequency with which they were expressed. Only one negative link (poverty -> learning material, in A) met this criterion, all other links are positive. Links of the generic structure are highlighted in red and shown in In (B), the links by which components affect themselves are not shown, for clarity, but particularly for the generic structure component were among the most widely reported (see Fig 3). Interest stands for child engagement in and motivation for education; attendance for child attendance, results for child learning outcomes, homework for child completion of homework, focus for child focus on learning, material for learning material, lesson for lesson learning, parent focus for parental focus on child learning.

components and the positive influence of each component on itself; Fig 3A). The frequency of these links were also seen within each of the group characteristics (S1–S5 Tables).

## Other feedback loops contiguous to the generic structure

To find what other system components were commonly expressed as connected to the *engagement-attendance-inclusion* loop, we investigated the other CLD components which most frequently affected the generic structure and were in turn affected by it. Because these contiguous loops may act to reinforce parts of the generic structure, they have the potential to effect accumulating increase or decrease in the generic structure components. We included all components with 1) a link to any component of the generic structure and 2) a link from any component of the generic structure, where each link was expressed by a majority (>50%) of the CLDs within any of our group characteristics (role, district, wealth, sex, time period).

This method identified only one common contiguous loop across characteristics: *child focus on learning*. This was expressed as a reinforcing loop contiguous with the generic structure commonly in both Afghanistan (51% frequency) and Pakistan (74% frequency), and in both period 1 (58% frequency) and period 2 (70% frequency) (Fig 4, Table 3).

Other system components were expressed as contiguous loops only by majorities of CLDs within groups that shared certain characteristics (i.e. role in the system, geography, sex, level of wealth, time period). The component *child learning outcomes* as a reinforcing loop contiguous with the generic structure was expressed by 70% of CLDs in Pakistan (Table 3). This loop was

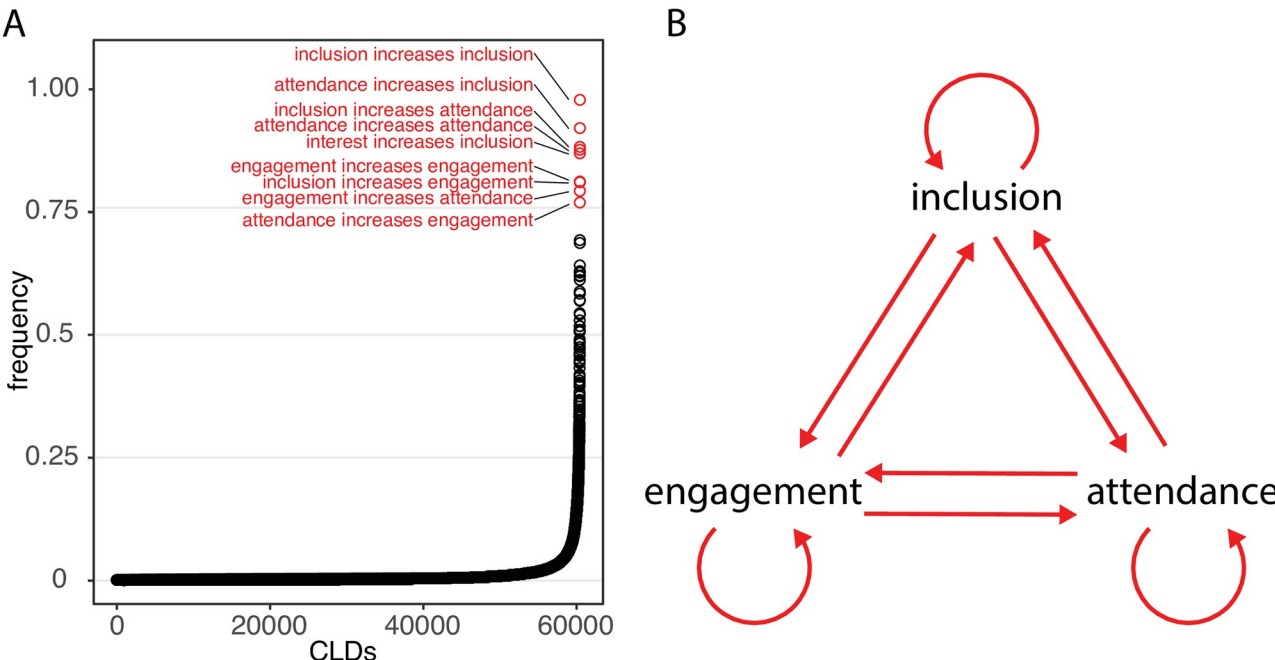

**Fig 3. A generic structure for education.** Note: (A) Across a wide range of local perceptions expressed in CLDs by participants across countries, languages, socioeconomic strata, and roles in the educational system, three components stood out: child interest and participation in education (*engagement*), child attendance (*attendance*), and child inclusion in equitable and quality education (*inclusion*). The reinforcing causal links among these components (arrows, with weight proportional to frequency with which causal link was reported) were the most widely reported of 60,419 causal links (B) and the only components expressed by >75% of CLDs.

also expressed in the majority of period 1 CLDs, and of CLDs built by students and parents. *Child homework completion* was also expressed as a reinforcing loop contiguous with the generic structure by >50% of CLDs in Pakistan, in one administrative area of Afghanistan, in period 2 and by students and teachers. Both *learning outcomes* and *homework completion* loops were expressed in the majority of CLDs built by participants associated with lowest-wealth 20% and middle-wealth 60% (Fig 4). The three components were expressed in a large majority of CLDs built by female.

Less frequently expressed contiguous loops included *parental involvement in child education* as a reinforcing loop, expressed by a majority of CLDs built by teachers; *teacher focus on child education* as a reinforcing loop, expressed by a majority of CLDs built in Pakistan; and *lesson learning* as a reinforcing loop, expressed by a majority of the lowest wealth group. These components were also expressed by majorities within certain districts, as were reinforcing contiguous loops around *child happiness and pride in learning*; *learning material* (books, pens, notebooks, uniforms); *parental encouragement of child learning*; *parental happiness and satisfaction*, *teacher happiness and satisfaction*, and *(decrease in) child focus on sports and games*. Balancing loops contiguous with the fixed structure were only reported by the majority of CLDs in Vehari, and involved the *learning outcomes, homework, lesson learning, parental involvement, teacher focus*, and *sports and games* components mentioned above.

## Characteristics that affect differences among perceptions of educational system

In addition to identifying a generic structure commonly reported across CLDs amid the diversity of casual links reported, we also employed multivariate tools borrowed from ecological

**Table 2. Components of the educational system with highest frequencies stratified by country, role in the education system, sex, wealth level and period of observation.**

| Strata | | System components | | | | | | |
|--------|--------|-----------|------------|------------|-------------------|-------------|----------|--------------|
| | | inclusion | attendance | engagement | learning material | child focus | homework | parent focus |
| overall | | 1 | 0.95 | 0.90 | 0.74 | 0.72 | 0.66 | 0.66 |
| country | Pakistan | - | 0.94 | 0.85 | 0.75 | 0.81 | 0.77 | 0.77 |
| | Afghanistan | - | 0.96 | 0.96 | 0.71 | 0.60 | 0.52 | 0.51 |
| role | teacher | - | 0.94 | 0.86 | 0.79 | 0.69 | 0.67 | 0.79 |
| | student | - | 0.96 | 0.88 | 0.72 | 0.78 | 0.70 | 0.51 |
| | parent | - | 0.93 | 0.92 | 0.73 | 0.72 | 0.67 | 0.74 |
| | SMC | - | 0.95 | 0.97 | 0.68 | 0.63 | 0.49 | 0.56 |
| Sex | Female group | | 0.93 | 0.86 | 0.79 | 0.73 | 0.72 | 0.78 |
| | male group | | 0.96 | 0.95 | 0.65 | 0.67 | 0.56 | 0.56 |
| | mixed group | | 0.96 | 0.89 | 0.79 | 0.72 | 0.72 | 0.63 |
| wealth | low | - | 0.94 | 0.88 | 0.74 | 0.72 | 0.81 | 0.75 |
| | mid | - | 0.93 | 0.89 | 0.72 | 0.76 | 0.64 | 0.66 |
| | high | - | 0.99 | 0.93 | 0.75 | 0.59 | 0.56 | 0.57 |
| period | pre- | - | 0.95 | 0.89 | 0.71 | 0.68 | 0.57 | 0.63 |
| | post- | - | 0.95 | 0.91 | 0.76 | 0.76 | 0.75 | 0.68 |

Note: Highest frequency components: inclusive and equitable quality education (inclusion, present in every CLD), child attendance (attendance), and child engagement in and motivation for education (engagement). There is less consensus on commonly cited components including learning material, child focus on learning (child focus), child homework completion (homework) and parental focus on child learning (parent focus). These overall patterns were also seen when stratifying CLDs built by participants in the two countries, by participants in different roles in the educational system, by groups of differing wealth categories, and pre- and post-implementation of a participant-designed intervention.

analysis to identify particular characteristics of model-building groups that significantly affect the similarity or difference in the CLDs they report, and to quantify how important these characteristics are to affecting these CLDs. Here, we found clear support for the importance of local conditions in shaping perceptions of the educational system: the characteristic with by far the most impact on CLDs composition and structure was the school at which the model was produced ($p<0.001$, $r^2 = 0.32$, Fig 5). In contrast, increasingly larger geographical areas showed lower fit, with $r^2 = 0.04$ for administrative area ($p<0.001$) and $r^2 = 0.01$ for country ($p<0.001$). Average rural wealth index of model-building group participants had a significant but low fit relationship with CLDs composition and structure ($r^2 = 0.01$, $p<0.01$), while average goods wealth index was not significant ($p = 0.39$). Sex, expressed as the percentage of a model-building group that was male participants, was marginally significant and low fit ($p = 0.04$, $r^2 = 0.01$). Other characteristics tested that did not show significant fit to CLD composition and structure were role of participants in the educational system ($p = 0.41$) and time period ($p = 0.98$).

## Discussion

We identified a generic reinforcing structure of education perceptions composed of three main components: *child engagement in education*, *child attendance*, and *child inclusion in equitable quality education*. We employed novel methods for comparative analysis of causal loop diagrams, an investigation that has been called for in the literature [53, 54].

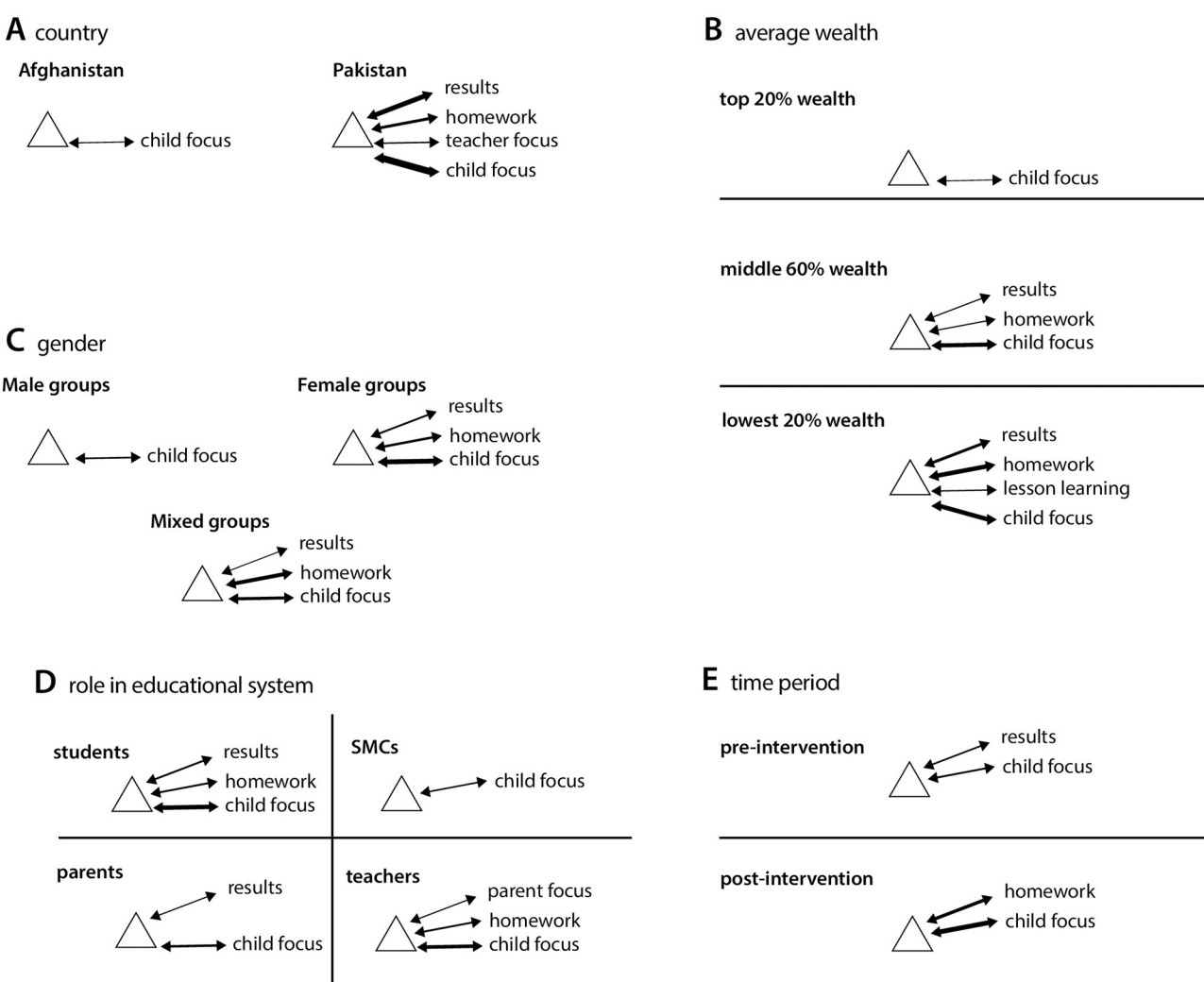

**Fig 4. Common reinforcing loops of the generic structure.** Note: (A) geography, (B) sex, (C) wealth category, (D) role in educational system, (E) time period. Components shown are those expressed in at least 50% of CLDs as a reinforcing loop of the generic structure (that is, both driving and being driven by it). Arrow line weights are proportional to frequency with which the loop was expressed by CLDs built by group participants in a country, wealth category, role, sex or time period. Results stands for child learning outcomes, homework for child completion of homework, child focus for child focus on learning.

## Child engagement in and motivation for education

Engagement is a complex construct with multiple definitions with some shared emphasis in the literature on students' active participation in learning activities and commitment to learn [57, 58]. Authors have identified different dimensions of engagement. Schaufeli and colleagues (2002) refer to a feeling of vigor in studying, dedication to learning and absorption in learning related activities [59–61]. The three dimensions refer respectively to persistence when learning activities are complicated, enthusiasm and pride in engaging with the material and finally a sense of deep concentration in studies. Fredricks et al. (2004) identified behavioral engagement characterized essentially by a positive conduct and participation in learning and academic-related tasks but also in activities related to school such as governance [62]. Fredricks et al. (2004) also referred to cognitive engagement which encompasses willingness to make the effort

**Table 3. Loops components with highest frequencies feeding into the generic structure.**

| Strata | | Loops components | | |
|---|---|---|---|---|
| | | child focus | child learning outcomes | Child homework completion |
| overall | | | | |
| country | Pakistan | 0.75 | 0.70 | 0.63 |
| | Afghanistan | 0.51 | 0.26 | 0.42 |
| role | teacher | 0.66 | 0.48 | 0.58 |
| | student | 0.69 | 0.58 | 0.58 |
| | parent | 0.61 | 0.52 | 0.48 |
| | SMC | 0.55 | 0.31 | 0.43 |
| sex | male group | 0.56 | 0.39 | 0.45 |
| | female group | 0.70 | 0.58 | 0.61 |
| | mixed group | 0.64 | 0.50 | 0.58 |
| wealth | Low 20% | 0.69 | 0.63 | 0.69 |
| | Mid 60% | 0.68 | 0.52 | 0.52 |
| | High 20% | 0.50 | 0.35 | 0.41 |
| period | pre-intervention | 0.58 | 0.56 | 0.41 |
| | post-intervention | 0.70 | 0.44 | 0.65 |

to understand difficult tasks, use adapted approach to learning and self-regulation to support learning. Finally, emotional engagement refers to positive and negative feeling towards school features, teachers, classmates, learning activities. Emotional engagement translates into interest, happiness and avoidance of boredom, anxiousness or sadness, overall disaffection from any kind of involvement into classroom activities.

Enthusiasm and pride in learning were often expressed by children: "When I come back home, I do my homework and prepare for the next day lesson" (Ghazni province) "Besides studying at home, we also do homework. But when I can, I like to read books, practice

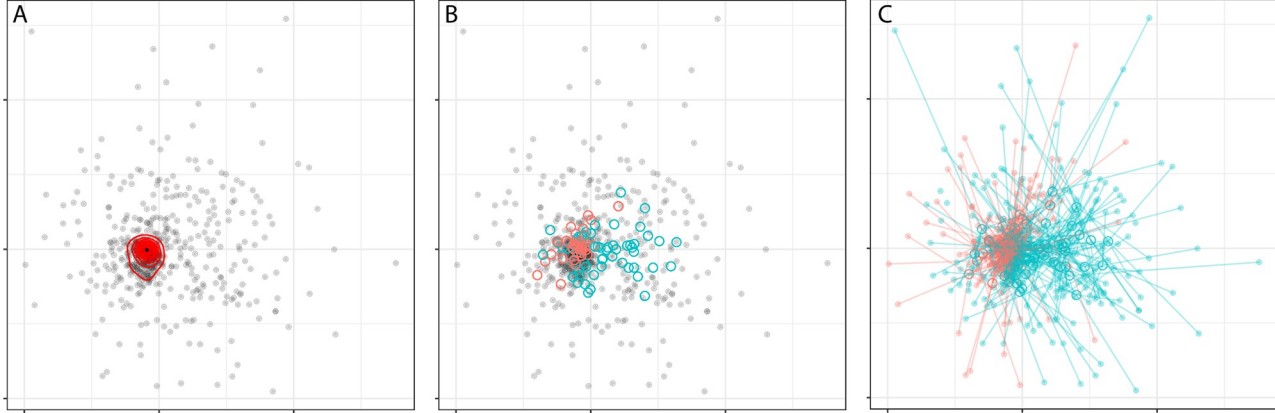

**Fig 5. Ordination of CLDs by non-metric multidimensional scaling.** Note: Each point is one CLD, with the distances among them determined by the difference in the causal loops that they express. The vast majority of CLDs (red contours in panel A) occupy a small space, showing the common generic structure they share. Despite this, different schools (circles in panel B, colored by country) occupy distinct points in the ordination space, showing the importance (r2 = 0.33) of local factors in determining CLDs composition and structure. However, much variation is not explained by geographic locality, and differences driven by the role in educational system of model builders, or the effects of the intervention, push some CLDs far away from the circle of their school (line segments in panel C).

drawings, do my homework" (Takhar). Children also expressed cognitive engagement as willingness to make effort: "Because some students don't do their homework, they cannot follow the next lesson; while those children who are doing their homework always understand the lesson" (Takhar). Emotional engagement was expressed for instance by students in Badakhshan: "when students lose interest in learning, they leave school. When they drop out, they don't learn anything, they stay illiterate and will remain shepherds or farmers in the future".

Such perspectives of engagement have been shown to be associated with academic performance [63, 64]. School stakeholders expressed this association between engagement and performance. For instance, students in a school of Rahim Yar Khan district argued that "*if students feel comfortable and relaxed, then it increases their interest in education and they learn better.*"

## Class attendance

Attendance is equally a central feature of learning. Being stressed and anxious is one among many reasons for skipping school, a major issue in both countries. Poverty, resulting in child labor and household chores [65] as well as family not being able to provide uniform, books and stationery is a source of anxiety and shame for children who prefer skipping school as a result. Finally, a school environment characterized by insecurity on the way to school, buildings in disrepair, scarcity of teaching material, lack of female teachers, and poorly trained, discriminatory, or violent teachers discourages students from coming to school [66–68].

Teachers in Vehari reported: "Children don't have notebooks, books, pens or pencils so they can't do their homework". They might skip school because "when children come to school without notebooks, pencils and bags, the teacher will scold them". Negative teachers' behavior in general decreases attendance and "positive attitude in class effectively reduce students' absenteeism" according to teachers in a Ghazni school. Assisting to class requires to overcome many other barriers. Teachers mentioned household chores: "What will happen if household chores of the children decrease? Children will be able to study more; they will come to school daily; they will complete their homework and focus on studies." (Rahim Yar Khan). Children are discouraged to come by the poor school environment: "The school doesn't have surrounding walls and so animals come and soil our school; besides, there is no clean drinking water source at school. Also, the classrooms don't have chairs and desks therefore we sit on the ground which is humid and dusty. The floor is not cemented, and walls are in raw materials. Additionally, there is no playground for the children to play during recess" (Takhar).

## Inclusive and equitable quality education

Inclusive and equitable quality education has been consistently considered in the literature as a necessary but not sufficient condition for effective learning [1, 15, 64, 69]. This approach was defined in multiples ways, in different schools, in particular as "all students taking part in learning", "all students being able to read and write", "all students being enrolled", "students coming to class everyday", "students paying attention in class".

These three components were reported to influence each other contributing to learning achievement and a successful learning experience independently from group characteristics. In both time periods (before and after an intervention composed of teacher training and collectively identified action ideas), we documented stable and universally shared causal relationships between child engagement, attendance and inclusion in equitable and quality education to be reported by majorities across all of localities, participants' roles in the educational system, sex, and qualitative and quantitative wealth differences. Prior studies, conducted mostly in high income countries contexts have suggested evidence of similar structure: teaching

practices fostering a protected environment conducive of the inclusion of *all* students, whatever their personal characteristics and the challenges they face, support students attendance [70], achievement [71] and engagement [72].

## Other factors interaction with the generic structure

We determined that a majority of workshop participants had converging views about a series of factors interacting with this generic structure. Child focusing their attention on learning was identified in most CLDs: "*If children pay attention and listen to the teacher, they will learn the lesson*" (Ghazni). Good results and homework completion appeared in a majority of CLDs in Pakistan and the latter in some areas of Afghanistan as well: "*When we learn our lesson then we see our results improving. We are then happy and proud. Being happy in class, and proud of our results, we focus more in class.*" (Children in Vehari).

All predicted better inclusion, engagement, and attendance and conversely the latter was reinforcing each of these factors. These findings suggest that an overall conducive environment expressed by the generic structure elicits child focus on learning and homework and good academic achievement that in return encourage attendance, engagement, and inclusion in an equitable and quality education system. Existing literature points to structural factors, such as supportive environment, structures and practices that are responsive to students learning and development needs [1], as well as to child psychological factors to predict academic achievement, but does not explicitly recognize the virtuous and reinforcing processes between structural and psychological factors considered together in a system of successful inclusion and equitable participation in a learning process of quality as shown in the present study.

## Role of the school environment

The importance of the school environment in shaping the composition and structure of the educational system expressed by participants was not surprising. A large body of literature emphasizes that student academic achievement cannot solely be driven by a generic structure or top-down policies but also relies on the arrangements of local resources (small classes, equipment), management style promoting a positive school climate and the teaching body role and practices (quality teaching, caring and respectful attitude, positive teachers-students relationships) [1, 72–75]. The relatively high fit of school suggests that *within* each school, students, teachers, parents, and SMC members perceived similar locally specific factors in their educational system, and that these factors were stable across time/intervention.

## Parents role in child education

Interestingly, teachers mentioned parents' involvement in their child education as an essential feature of a successful educational system. Teachers noted that "*poor parents cannot encourage their children to learn because they are busy working for survival*"; they argued as well that "*if parents fight with each other, their children cannot focus on studies because they are stressed and anxious*"; furthermore, they found that "*parents lack of attention to their children's education discourage them and it shows in overlooked homework, lower attendance and wicked results*". Conversely "*if parents do not neglect their children, they give more time to their studies*" (Vehari). Indeed, parents, and family more generally, have long been considered to provide the means and the caring environment for better learning outcomes [76] and reduction in detrimental behaviors (truancy, drop out) [77], particularly if they accompany child learning effort and greater expectations eliciting stronger engagement [78]. Parents' support can help students meet expectations in terms of learning, attendance, and engagement. It may also shield against the effect of external stressors (poverty, conflict) and their negative

consequences for child mental wellbeing and functioning by promoting resilience [79]. Our findings from many CLDs confirm the role of parental encouragement in the child effort to learn. They also hint at the importance of their own happiness and satisfaction when the child is effectively learning, triggering a virtuous cycle of more parental satisfaction creates more child engagement and vice versa. In a Gothki district school parents noted that "*When the child participation in class triggers his/her interest about learning, teachers feel encouraged and happy leading to our own happiness and satisfaction. We will then be more willing to provide the needed material to study. We also noted that teachers' scolding tends to lessen while child attendance and participation in class tend to surge*". Of note, parental support and satisfaction were associated with the overall generic structure beyond the sole factor of child engagement.

## Teachers' role in the learning process

A majority of participants in Pakistan argued that teachers' role was also instrumental in child engagement, attendance and inclusion. Teachers' encouragement was viewed as essential by wealthier participants. Supportive teacher-student relationships are associated to child performance and engagement [73]. Children from a school in Rahim Yar Khan stated "*when our teacher shows concern and warmth, we feel happy and proud to learn, we are less anxious and afraid of our teacher. We take more interest in learning and our results improve*". It is noteworthy that the only balancing causal feedback loops comprise both *parents' involvement in their child education* and *teacher's focus on their students* as factors increasing *homework completion* and *lessons learning* but diminishing *child focus on sports and* (particularly) *games* instead of learning.

Our findings also identify the role of a dual level system of support involving teachers and parents. Securing effective inclusion of all children by focusing on universal attendance and active student engagement resonates with the principles of the education for all framework [15] and the fourth sustainable development goal (SDG 4) which aims at "ensur[ing] inclusive and equitable quality education [. . .] for all" [17].

## Role of community stakeholders

In practice, it is of note that the school community driven interventions (three to five action ideas agreed to implement to improve inclusion, equity, and quality of education) did not affect the generic structure of the system. Yet, taking into consideration the generic structure, decisions taken consensually by school actors represented in school management committees may be instrumental in improving students learning experience and outcomes. Even with a stable system, interventions may drive feedback loops towards desired outcomes. In ongoing work, we study the learning outcomes of the interventions and hope that this will shed light on their effectiveness to supporting academic achievement and life skills.

## Policy implications

Our findings have important policy, research, and practical implications. The generic structure combines structural and contextual factors (an environment encouraging attendance, parents, and teachers' engagement), psychological or internal factors (child engagement in and focus on learning) with academic achievement (child inclusion and participation in learning). Considering this generic structure will avoid the manifestation of preventable difficulties, such as the negative impact of school non-attendance on multiple outcomes (distress and anxiety, crime, substance abuse, unwanted pregnancy, drop out of school, training or employment, and poverty) [80–82] or of lack of inclusion on attendance, engagement and eventually children's results.

This constitutes a first step towards reconciling diverse views about what inclusive and equitable quality education encompasses. Any educational reform should simultaneously embrace learners' diversity, combine policy principles of ensuring easy equitable access to foster attendance, promote student interest in learning and meet their needs through child centered pedagogy and non-discriminatory teaching practices while empowering school communities to implement those principles. Future research should investigate further, in other contexts, the common ground expressed by school communities across the board in Afghanistan and Pakistan.

## Supporting information

**S1 Table. Main links by role in the education system.**
(CSV)

**S2 Table. Main links by geographical contexts.**
(CSV)

**S3 Table. Main links by sex.**
(CSV)

**S4 Table. Main links by wealth.**
(CSV)

**S5 Table. Main links by time periods.**
(CSV)

## Acknowledgments

We are thankful for feedback from Elaine Unteralter, Steve Taft and Ramesh Raghavan on previous versions of this article. We are also thankful to our colleagues in Afghanistan and Pakistan who conducted the group model building workshops as well as to the many students, teachers, parents and school management committee members who participated in these workshops and offered their views. This paper is dedicated to the memory of Parween Azimi and Munib Sohail, whose contributions were central to the success of this study.

## Author Contributions

**Conceptualization:** Jean-Francois Trani.

**Data curation:** Robbie Hart.

**Formal analysis:** Robbie Hart.

**Funding acquisition:** Jean-Francois Trani.

**Investigation:** Jean-Francois Trani.

**Methodology:** Jean-Francois Trani.

**Project administration:** Jean-Francois Trani.

**Resources:** Jean-Francois Trani.

**Supervision:** Jean-Francois Trani.

**Writing – original draft:** Jean-Francois Trani.

**Writing – review & editing:** Robbie Hart.

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
