## [Decision Letter · Decision Letter 0]

13 Mar 2023

PONE-D-23-00992Student engagement, attendance and inclusion in the learning process are central mechanisms determining academic achievement: Evidence from Afghanistan and PakistanPLOS ONE

Dear Dr. Trani,

Thank you for submitting your manuscript to PLOS ONE. After careful consideration, we feel that it has merit but does not fully meet PLOS ONE’s publication criteria as it currently stands. Therefore, we invite you to submit a revised version of the manuscript that addresses the points raised during the review process.

We look forward to receiving your revised manuscript.

Kind regards,

Grant Rich, Ph.D.

Academic Editor

PLOS ONE

Journal Requirements:

2. Please provide additional details regarding ethical approval in the body of your manuscript. In the Methods section, please ensure that you have specified the name of the IRB/ethics committee that approved your study.

3. Please provide additional details regarding participant consent. In the Methods section, please ensure that you have specified (1) whether consent was informed and (2) what type you obtained (for instance, written or verbal). If your study included minors, state whether you obtained consent from parents or guardians. If the need for consent was waived by the ethics committee, please include this information.

4. Please include a complete copy of PLOS’ questionnaire on inclusivity in global research in your revised manuscript. Our policy for research in this area aims to improve transparency in the reporting of research performed outside of researchers’ own country or community. The policy applies to researchers who have travelled to a different country to conduct research, research with Indigenous populations or their lands, and research on cultural artefacts. The questionnaire can also be requested at the journal’s discretion for any other submissions, even if these conditions are not met.  Please find more information on the policy and a link to download a blank copy of the questionnaire here: https://journals.plos.org/plosone/s/best-practices-in-research-reporting. Please upload a completed version of your questionnaire as Supporting Information when you resubmit your manuscript.

8. We note that Figure 4 in your submission contain map images which may be copyrighted. All PLOS content is published under the Creative Commons Attribution License (CC BY 4.0), which means that the manuscript, images, and Supporting Information files will be freely available online, and any third party is permitted to access, download, copy, distribute, and use these materials in any way, even commercially, with proper attribution. For these reasons, we cannot publish previously copyrighted maps or satellite images created using proprietary data, such as Google software (Google Maps, Street View, and Earth). For more information, see our copyright guidelines: http://journals.plos.org/plosone/s/licenses-and-copyright.

 a. You may seek permission from the original copyright holder of Figure 4 to publish the content specifically under the CC BY 4.0 license. 

9. We note that Figure 4 includes an image of a participant in the study.

Additional Editor Comments:

The population and topic of focus are significant- there is a paucity- especially in English of data/research on education in these two nations.

Improvements need to be made in the presentation to better communicate findings and to address reviewer feedback.

One reviewer voted to reject but the second reviewer asked for a minor revision, so the overall decision is to ask you to write and submit a "major revision"

Please read the reviewer feedback below and address their comments in your revision. You should especially be sure to address sex differences, and also reasons there may be sex differences, such as religion or culture or history or politics (for instance in India's Haryana state next to India's portion of Punjab, it is very Hindu but also has sex differences such as in proportion of male vs female births.

Be sure to work to clarify and more effectively communicate and explain your theory and your methods and results - the aim of research is to be able to communicate the information so that readers can follow the story you tell.

see reviewer comments below

Grant J Rich, PhD

PLOS ONE Student Engagement Afghanistan and Pakistan

REVIEWER ONE VOTE REJECT AND WROTE

the authors report that there is no consensus as to how quality education can be defined or indeed what the components of such an education could be and then proceed to conduct a very large study which investigates the components which go to child inclusion in the learning process. While child inclusion in the learning process is a noble concern, it could not be said to be the single defining characteristic of quality education. The study was conducted in Afghanistan and Pakistan and the authors do not seem to consider sex differences between male and female children which would definitely impact on the results of the study. I found the manuscript to be really dense to read. The results section was almost impossible to decipher. While child results were found (Table 2) to be a loops component (along with child focus and homework) at least for some of the variables investigated, it is not clear to me that the title indicating the three mechanisms which are central in determining academic achievement is accurate. So the main message the authors wish to convey is that child engagement, child attendance, and child inclusion determine, inter alia, child results. This seems to me to be an overgeneralisation of the results and indeed is not something which most people would consider new knowledge. Generally the manuscript was well written, however there are some grammar issues. There are also instances of in-words being used. As an academic working in the teaching and learning area, although not knowledgeable in the statistics and method used in this study, I found the manuscript almost incomprehensible in parts. If the idea of dissemination via a journal article is dissemination, then the language used should be more easily understood. The logical flow of the manuscript was also lacking in parts. If accepted, the manuscript should be carefully proofread for grammar and/or typing errors and for the use, undefined, of in-house language. The authors say the data are fully available without restriction, but I could not see where the data were available from other than the UK data service which is a controlled data service and you have to apply for access.(less...)

REVIEW TWO VOTED MINOR REVISION AND WROTE

This manuscript on student engagement, motivation, and inclusion deals with information with which I have relatively superficial knowledge. As such, I hope my comments provide useful information for the authors. Overall, I think this manuscript will be an important contribution to the area.

The literature review seems quite thorough and lays out the important issues associated with the research question, and the methodology seems sound. Similarly, the results seem well structured, although I am not familiar with in-depth details about the analysis the researchers conducted. Nonetheless, the graphic presentation of the data, although complex, provides the important detail needed to make sense of the results.

The only aspects that I have comments about follow. They are fairly minor.

1. The authors mention causal mechanisms regarding the variables they identified. I assume it is perceived causation, not causation as produced by experimental methods. As an outsider to this field, I don't know the customs in discussing such aspects of the results, but the element of perception in assessing causal connections seems important to me.

2. In the results, there is a single negative causal link (with multiple positive links). This seems curious, given that negative elements would seem to be related to elements of motivation and engagement and inclusion. Is there a potential explanation for this? Is it an artifact of the nature of the data collected?

3. In the discussion, I was distracted by the quotations. It seems to me that it would be helpful to establish the central points of discussion in a specific and concrete way, with subsequent presentation of the quotations. As it was, I had to move through many statements in an attempt to connect the important ideas.

4. One more theoretical thought I had concerns the degree to which the factors and components are under the control of the various stakeholders. I think it would be an important addition to discuss not only the important components but also the degree to which students, parents, and teachers can structure the environment for optimal outcomes

Reviewers' comments:

Reviewer's Responses to Questions

**Comments to the Author**

1. Is the manuscript technically sound, and do the data support the conclusions?

Reviewer #1: Partly

Reviewer #2: Yes

2. Has the statistical analysis been performed appropriately and rigorously? 

Reviewer #1: I Don't Know

Reviewer #2: Yes

3. Have the authors made all data underlying the findings in their manuscript fully available?

Reviewer #1: No

Reviewer #2: Yes

4. Is the manuscript presented in an intelligible fashion and written in standard English?

Reviewer #1: No

Reviewer #2: Yes

5. Review Comments to the Author

Reviewer #1: The authors report that there is no consensus as to how quality education can be defined or indeed what the components of such an education could be and then proceed to conduct a very large study which investigates the components which go to child inclusion in the learning process. While child inclusion in the learning process is a noble concern, it could not be said to be the single defining characteristic of quality education. The study was conducted in Afghanistan and Pakistan and the authors do not seem to consider sex differences between male and female children which would definitely impact on the results of the study. I found the manuscript to be really dense to read. The results section was almost impossible to decipher. While child results were found (Table 2) to be a loops component (along with child focus and homework) at least for some of the variables investigated, it is not clear to me that the title indicating the three mechanisms which are central in determining academic achievement is accurate. So the main message the authors wish to convey is that child engagement, child attendance, and child inclusion determine, inter alia, child results. This seems to me to be an overgeneralisation of the results and indeed is not something which most people would consider new knowledge. Generally the manuscript was well written, however there are some grammar issues. There are also instances of in-words being used. As an academic working in the teaching and learning area, although not knowledgeable in the statistics and method used in this study, I found the manuscript almost incomprehensible in parts. If the idea of dissemination via a journal article is dissemination, then the language used should be more easily understood. The logical flow of the manuscript was also lacking in parts. If accepted, the manuscript should be carefully proofread for grammar and/or typing errors and for the use, undefined, of in-house language. The authors say the data are fully available without restriction, but I could not see where the data were available from other than the UK data service which is a controlled data service and you have to apply for access.

Reviewer #2: This manuscript on student engagement, motivation, and inclusion deals with information with which I have relatively superficial knowledge. As such, I hope my comments provide useful information for the authors. Overall, I think this manuscript will be an important contribution to the area.

The literature review seems quite thorough and lays out the important issues associated with the research question, and the methodology seems sound. Similarly, the results seem well structured, although I am not familiar with in-depth details about the analysis the researchers conducted. Nonetheless, the graphic presentation of the data, although complex, provides the important detail needed to make sense of the results.

The only aspects that I have comments about follow. They are fairly minor.

1. The authors mention causal mechanisms regarding the variables they identified. I assume it is perceived causation, not causation as produced by experimental methods. As an outsider to this field, I don't know the customs in discussing such aspects of the results, but the element of perception in assessing causal connections seems important to me.

2. In the results, there is a single negative causal link (with multiple positive links). This seems curious, given that negative elements would seem to be related to elements of motivation and engagement and inclusion. Is there a potential explanation for this? Is it an artifact of the nature of the data collected?

3. In the discussion, I was distracted by the quotations. It seems to me that it would be helpful to establish the central points of discussion in a specific and concrete way, with subsequent presentation of the quotations. As it was, I had to move through many statements in an attempt to connect the important ideas.

4. One more theoretical thought I had concerns the degree to which the factors and components are under the control of the various stakeholders. I think it would be an important addition to discuss not only the important components but also the degree to which students, parents, and teachers can structure the environment for optimal outcomes.

6. PLOS authors have the option to publish the peer review history of their article (what does this mean?). If published, this will include your full peer review and any attached files.

Reviewer #1: No

Reviewer #2: **Yes: **Bernard C. Beins

Grant J. Rich, PhD LMT BCTMBPresident-Elect Society for Peace, Conflict, and Violence (APA D48)President-Elect Society for Media Psychology and Technology (APA D46)

Fellow, Association for Psychological Science (APS)Fellow, American Psychological Association (APA)Senior Contributing Faculty, Walden UniversityDr. Rich's SPN Website: http://rich.socialpsychology.org/**Book Website** (Rich, Gielen, & Takooshian, 2017)http://www.infoagepub.com/products/Internationalizing-the-Teaching-of-Psychology**Book Website** (Rich & Sirikantraporn, 2018)https://rowman.com/ISBN/9781498554831/Human-Strengths-and-Resilience-Cross-Cultural-and-International-Perspectives#**Book Website** (Rich, Jaafar, & Barron, 2020) Psychology in Southeast Asia. Routledge.https://www.routledge.com/Psychology-in-Southeast-Asia-Sociocultural-Clinical-and-Health-Perspectives/Rich-Jaafar-Barron/p/book/9780367492144**Book Website** (Rich & Ramkumar, 2022) Psychology in Oceania and the Caribbean, Springerhttps://link.springer.com/book/10.1007/978-3-030-87763-7#editorsandaffiliations **Book Website**(Rich, Kuriansky, Gielen, & Kaplan, in press) * Psychosocial Experiences and Adjustment of Migrants: Coming to the USA, Elsevier**https://www.elsevier.com/books/psychosocial-experiences-and-adjustment-of-migrants/rich/978-0-12-823794-6* **Book **(Rich, Kumar, & Farley, in contract)* Handbook of **Media Psychology and Technology-The Science and the Practice,** Springer*

---

## [Author Response · Author response to Decision Letter 0]

31 Jul 2023

Editor Comments:

Comment 1. Please ensure that your manuscript meets PLOS ONE's style requirements, including those for file naming. 

Response to Comment 1. We revised our manuscript to meet PLOS ONE's style requirements. Titles, reference style, tables, figures and file naming have been changed to this effect.

Comment 2. Please provide additional details regarding ethical approval in the body of your manuscript. In the Methods section, please ensure that you have specified the name of the IRB/ethics committee that approved your study.

Response to comment 2: We modified the Methods section of the manuscript to add the name of the IRB/ethics committee that approved our study. 

The text reads as follows (See p. 11 l. 249-253):

“Ethical clearance was received from the Ministry of Education in Afghanistan, the National Rural Support Program Board in Pakistan and Washington University in St Louis, Human Research and Protection Office. We obtained informed verbal consent from participants in group model building workshops because no personal information was collected during the workshop.”

Comment 3. Please provide additional details regarding participant consent. In the Methods section, please ensure that you have specified (1) whether consent was informed and (2) what type you obtained (for instance, written or verbal). If your study included minors, state whether you obtained consent from parents or guardians. If the need for consent was waived by the ethics committee, please include this information.

Response to comment 3: We modified the Methods section of the manuscript to add the requested information about consent:

The text reads as follows (See p. 11 l. 249-253):

“Ethical clearance was received from the Ministry of Education in Afghanistan, the National Rural Support Program Board in Pakistan and Washington University in St Louis, Human Research and Protection Office. We obtained informed verbal consent from participants in group model building workshops because no personal information was collected during the workshop.”

Comment 4. Please include a complete copy of PLOS’ questionnaire on inclusivity in global research in your revised manuscript.

Response to Comment 4: A complete version of PLOS’ questionnaire on inclusivity in global research has been added as Supporting Information our revised manuscript.

Comment 5. In your Data Availability statement, you have not specified where the minimal data set underlying the results described in your manuscript can be found….

Response to comment 5: The study study's minimal data set will be made available on UK data repository https://ukdataservice.ac.uk/ The database is fully available upon request to the UK data repository service.

Comment 6. We note that you have stated that you will provide repository information for your data at acceptance. Should your manuscript be accepted for publication, we will hold it until you provide the relevant accession numbers or DOIs necessary to access your data. If you wish to make changes to your Data Availability statement, please describe these changes in your cover letter and we will update your Data Availability statement to reflect the information you provide.

Response to comment 6: As soon as the paper is accepted we will provide the DOI from UK data repository service (https://ukdataservice.ac.uk/)

Comment 7. Please include your full ethics statement in the ‘Methods’ section of your manuscript file. In your statement, please include the full name of the IRB or ethics committee who approved or waived your study, as well as whether or not you obtained informed written or verbal consent. If consent was waived for your study, please include this information in your statement as well.

Response to comment 7: We modified the ‘Methods’ section of the manuscript to add the requested information about consent:

The text reads as follows (See p. 11 l. 249-253):

“Ethical clearance was received from the Ministry of Education in Afghanistan, the National Rural Support Program Board in Pakistan and Washington University in St Louis, Human Research and Protection Office. We obtained informed verbal consent from participants in group model building workshops because no personal information was collected during the workshop.”

Comment 8. We note that Figure 4 in your submission contain map images which may be copyrighted…

Response to comment 8: We removed the map images from this figure.

Comment 9. We note that Figure 4 includes an image of a participant in the study.

Response to comment 8: We removed the participant images from this figure.

Additional Editor Comments:

Comment 1. The population and topic of focus are significant- there is a paucity- especially in English of data/research on education in these two nations.

Improvements need to be made in the presentation to better communicate findings and to address reviewer feedback. Be sure to work to clarify and more effectively communicate and explain your theory and your methods and results - the aim of research is to be able to communicate the information so that readers can follow the story you tell.

Response to comment 1. Thank you for indicating that the topic is of interest. We agree with the editor and the reviewer that it is important for scientists to convey effectively and clearly the scope, the theory, the methods and the results of their study. We have therefore comprehensively revised the manuscript (including title, abstract, and body) to add structure, reduce and standardize terminology, and increase clarity. In particular, we explained that the scope of the study was to examine if there are common factors that define “inclusive and equitable quality education”, describe how they are articulated with one another, and identify what are the best interventions to foster inclusive and equitable quality education. We kept all modifications in track change. We hope that the article reads now better. 

Comment 2. You should especially be sure to address sex differences, and also reasons there may be sex differences, such as religion or culture or history or politics (for instance in India's Haryana state next to India's portion of Punjab, it is very Hindu but also has sex differences such as in proportion of male vs female births.

Response to comment 2. Thank you for pointing out this important characteristic. Because many models were built by mixed groups, we did not originally analyze gender differences. In response to this comment, we’ve added the sex ratio of the model-building groups to our set of analytic characteristics, and find a significant but very small effect of sex ratio on causal loop diagram structure. Importantly, we found that the generic structure was similarly expressed by groups that were predominantly, male, predominantly female, or mixed-sex.

REVIEWER ONE VOTE REJECT AND WROTE

Comment 1. The authors report that there is no consensus as to how quality education can be defined or indeed what the components of such an education could be and then proceed to conduct a very large study which investigates the components which go to child inclusion in the learning process. While child inclusion in the learning process is a noble concern, it could not be said to be the single defining characteristic of quality education.

Response to comment 1. We agree with the reviewer that quality education is not defined uniquely by child inclusion in the learning process. To avoid any misunderstanding, we referred to the sustainable development goal 4 that mentions “inclusive and equitable quality education” for all as a goal shared by the United Nations general Assembly. We reviewed in a revised version of the introduction what the literature is arguing this goal entails. We hoped we argued effectively that quality education is not defined only by child inclusion in the learning process but that both require a set of conditions around the teacher, the school and its environment to be achieved. We also made clear from the beginning that the focus of our study is “inclusive and equitable quality education” not child results. 

Comment 2. The study was conducted in Afghanistan and Pakistan and the authors do not seem to consider sex differences between male and female children which would definitely impact on the results of the study. 

Response to comment 2. As mentioned already in response to the editor comment and in response to the present comment, we’ve added the sex ratio of the model-building groups to our set of analytic characteristics, and find a significant but very small effect of sex ratio on causal loop diagram structure. Importantly, we found that the generic structure was similarly expressed by groups that were predominantly, male, predominantly female, or mixed-sex. This is an important finding that reinforces the finding of a generic structure of quality education that goes beyond multiple demographic and socioeconomic characteristics of interest. 

Comment 3. I found the manuscript to be really dense to read. The results section was almost impossible to decipher. 

Response to comment 3. We agree that the manuscript was dense, and we rewrote it entirely, including the ‘results’ section to clarify the argument. We hope the new shorter version reads well.

Comment 4. While child results were found (Table 2) to be a loops component (along with child focus and homework) at least for some of the variables investigated, it is not clear to me that the title indicating the three mechanisms which are central in determining academic achievement is accurate. So the main message the authors wish to convey is that child engagement, child attendance, and child inclusion determine, inter alia, child results. This seems to me to be an overgeneralisation of the results and indeed is not something which most people would consider new knowledge.

Response to comment 4. We clarified the title by referring to the sustainable development goal 4. The title reads now as follows: “Student engagement and attendance are central mechanisms interacting with inclusive and equitable quality education: Evidence from Afghanistan and Pakistan.” We hope the title as it is formulated now more clearly and accurately reflects the generic structure we identified through the analysis of the 648 causal loop diagrams. We also clarified how the three-mechanism (i.e. child attendance, child participation in the learning process and inclusive and equitable quality education) structure identified in 75% of causal loop diagrams works through reinforcing mechanisms to improve inclusive and equitable quality education. Yet, we also argued that other important factors interact with this generic structure such as child focus on learning, child learning outcomes, child homework completion among others. Therefore, we would argue here that our findings are new knowledge and that the generalization does not go beyond our findings for Afghanistan and Pakistan.

Comment 5. Generally the manuscript was well written, however there are some grammar issues. There are also instances of in-words being used. As an academic working in the teaching and learning area, although not knowledgeable in the statistics and method used in this study, I found the manuscript almost incomprehensible in parts. If the idea of dissemination via a journal article is dissemination, then the language used should be more easily understood.

The logical flow of the manuscript was also lacking in parts. If accepted, the manuscript should be carefully proofread for grammar and/or typing errors and for the use, undefined, of in-house language.

Response to comment 5. As indicated in response to comment 3, we agree with the reviewer and we have given the previous version of the manuscript a careful edit, with attention to order, clarity, and definition of all terms and concepts, while simplifying the language as much as possible. We are hopeful we removed all grammar and typing mistakes. 

Comment 6. The authors say the data are fully available without restriction, but I could not see where the data were available from other than the UK data service which is a controlled data service and you have to apply for access.(less...)

Response to comment 6. The data is on free access upon request to the UK data service website https://ukdataservice.ac.uk/. We also added supplementary data with the new submission of the manuscript to make more data immediately available. The full data is accessible at this repository, which is the official one requested by the main funding agency. The agency does not put further limitation on access beyond making a request.

REVIEW TWO VOTED MINOR REVISION AND WROTE

This manuscript on student engagement, motivation, and inclusion deals with information with which I have relatively superficial knowledge. As such, I hope my comments provide useful information for the authors. Overall, I think this manuscript will be an important contribution to the area.

The literature review seems quite thorough and lays out the important issues associated with the research question, and the methodology seems sound. Similarly, the results seem well structured, although I am not familiar with in-depth details about the analysis the researchers conducted. Nonetheless, the graphic presentation of the data, although complex, provides the important detail needed to make sense of the results.

The only aspects that I have comments about follow. They are fairly minor.

Comment 1. The authors mention causal mechanisms regarding the variables they identified. I assume it is perceived causation, not causation as produced by experimental methods. As an outsider to this field, I don't know the customs in discussing such aspects of the results, but the element of perception in assessing causal connections seems important to me.

Response to comment 1. We are very thankful for the reviewer’s positive comment and support for publication provided we address their comments. The reviewer is making an important point about “perceived causation” and the reviewer’s interpretation is correct. As part of a comprehensive revision, we have particularly emphasized that this is a study of stakeholders’ perceptions, and argued for why this is an important and relevant corpus to consider.

Comment 2. In the results, there is a single negative causal link (with multiple positive links). This seems curious, given that negative elements would seem to be related to elements of motivation and engagement and inclusion. Is there a potential explanation for this? Is it an artifact of the nature of the data collected?

Response to comment 2. We understand that most readers will be unfamiliar with the language of systems and that system dynamics uses a very specific conventional approach. More specifically, group model building and their outputs, causal loop diagrams, have specific conventions around the use of the plus and minus signs. We modified our manuscript to significantly add to the current explanation of the method. It must be argued that participants identified both positive and negative connections between factors. But it appeared that the most prevalent connections were part of reinforcing loops that usually have positive connections. We rewrote the principles of system dynamics as follows (p. 8, l. 176 to p. 9, l. 200):

“CLDs apply simple rules. They build on causal relationships between two components (a cause and an effect). For example, many groups expressed that if child attendance increases, child participation in the learning process increases as a result (see Fig 1). The reverse is implied: if child attendance decreases (as was observed during the COVID19 pandemic), child participation in the learning process decreases. This is an example of a positive causal relationship where an increase in the cause drives an increase in the effect (indicated during model building by arrows with a plus sign); negative causal relationships are also possible (indicated during model building by arrows with a minus sign) when an increase in the cause drives a decrease in the effect. For instance, if child attendance increases, then teacher scolding of children decreases. Causal relationships may be directly expressed (cause increases effect), or downstream relationships may be implied by the system (cause increases effect 1 which in turn increases effect 2). For example, the direct relationships i) child health increases child attendance and ii) child attendance increases inclusion in the learning process imply the downstream relationship iii) child health increases inclusion in the learning process, with child attendance as an intermediary component.

Because system thinking analyses the world through causal relationships in both directions, downstream causal relationships in a fully elaborated CLD will eventually form loops (cause increases effect 1…increases effect n increases cause). There are two possible types of loops: reinforcing and balancing. In reinforcing loops (also called vicious or virtuous cycles) an initial increase in a cause will drive downstream increase in that cause and an initial decrease will drive downstream decrease, acting to amplify change in the system. In balancing loops, an initial increase in a cause will drive downstream decrease in that cause (and vice versa) acting to counteract change in the system [52].”

Comment 3. In the discussion, I was distracted by the quotations. It seems to me that it would be helpful to establish the central points of discussion in a specific and concrete way, with subsequent presentation of the quotations. As it was, I had to move through many statements in an attempt to connect the important ideas.

Response to comment 3. We rewrote the discussion as suggested by the reviewer. We first mentioned the main argument before presenting the quotations as an illustration of the argument. We also introduced subsection titles in the discussion to guide the reader: child engagement in and motivation for education, class attendance, inclusive and equitable quality education, other factors interaction with the generic structure, role of the school environment, parents’ role in child education, teachers role in the learning process, role of community stakeholders and policy implications.

Comment 4. One more theoretical thought I had concerns the degree to which the factors and components are under the control of the various stakeholders. I think it would be an important addition to discuss not only the important components but also the degree to which students, parents, and teachers can structure the environment for optimal outcomes.

Response to comment 4. This is an important point, and we are grateful to the reviewer for bringing it to our attention. The present paper is part of a wider research study that used community-based system dynamics to engage with school stakeholders in defining their school system and what actions could be taken to improve the system. Such decisions were taken by consensus and non-governmental organizations who partnered in the study committed resources to help implement those decisions. Some of these actions were within the reach of these stakeholders, such as parents/teachers regular meeting, positive discipline, child participation in the learning process and did not require much support. Other actions required the intervention of external support and the NGOs partners provided material and equipment when a classroom needed to be built, but the village community would provide the land and the labor and took the lead in the implementation. This part of the research which had an impact on children learning outcomes is not published yet. We revised the discussion to add a paragraph explaining what role community stakeholders potentially can play to leverage the system (see p. 28 l. 720-729):

“Role of community stakeholders

In practice, it is of note that the school community driven interventions (three to five action ideas agreed to implement to improve inclusion, equity, and quality of education) did not affect the generic structure of the system. Yet, taking into consideration the generic structure, decisions taken consensually by school actors represented in school management committees may be instrumental in improving students learning experience and outcomes. Even with a stable system, interventions may drive feedback loops towards desired outcomes. In ongoing work, we study the learning outcomes of the interventions and hope that this will shed light on their effectiveness to supporting academic achievement and life skills.”

---

## [Editor Report · Decision Letter 1]

9 Aug 2023

Student engagement and attendance are central mechanisms interacting with inclusive and equitable quality education: Evidence from Afghanistan and Pakistan

PONE-D-23-00992R1

Dear Dr. Trani

We’re pleased to inform you that your manuscript has been judged scientifically suitable for publication and will be formally accepted for publication once it meets all outstanding technical requirements.

Kind regards,

Grant Rich, Ph.D.

Academic Editor

PLOS ONE

Additional Editor Comments (optional):

The authors took care to attend to the reviewer feedback and the article topic and population will make a significant contribution to our understanding of processes in these understudied regions

Reviewers' comments:

Grant J. Rich, PhD 

Candidate for President-Elect for the American Psychological Association

President-Elect Society for Peace, Conflict, and Violence (APA)

President-Elect Society for Media Psychology and Technology (APA)

Fellow, Association for Psychological Science (APS)

Fellow, American Psychological Association (APA) (D1, D2, D46, D48, D52)

Senior Contributing Faculty, Walden University, Juneau, Alaska

Editorial Board Member: PLOS ONE, APA's Peace & Conflict,
APA's Traumatology

Book Series Co-Editor w/ Anthony Marsella (U. Hawai'i), Springer. International and Cultural Psychology (ICUP)       

 https://www.springer.com/series/6089

Select Recent Books

(Rich, Gielen, & Takooshian,
2017).
Internationalizing the Teaching of Psychology.

IAP.

(Rich & Sirikantraporn, 2018).
Human Strengths and Resilience: Cross Cultural and International Perspectives. Rowman & Littlefield.

(Rich, Jaafar, & Barron, 2020).
Psychology in Southeast Asia. Routledge.

(Rich & Ramkumar, 2022).
Psychology in Oceania and the Caribbean. Springer. 

(Rich, Kuriansky, Gielen, & Kaplan, 2023) .
Psychosocial Experiences and Adjustment of Migrants: Coming to the USA. 
Elsevier.

(Rich, Kumar, & Farley, in contract).
Handbook of Media Psychology and Technology-The Science and the Practice. Springer.

---

## [Editor Report · Acceptance letter]

15 Sep 2023

PONE-D-23-00992R1 

Student engagement and attendance are central mechanisms interacting with inclusive and equitable quality education: Evidence from Afghanistan and Pakistan 

Dear Dr. Trani:

I'm pleased to inform you that your manuscript has been deemed suitable for publication in PLOS ONE. Congratulations! Your manuscript is now with our production department. 

Kind regards, 

on behalf of

Dr. Grant Rich 

Academic Editor

PLOS ONE